# Nanopore-Based Direct RNA Sequencing of the *Trypanosoma brucei* Transcriptome Identifies Novel lncRNAs

**DOI:** 10.3390/genes14030610

**Published:** 2023-02-28

**Authors:** Elisabeth Kruse, H. Ulrich Göringer

**Affiliations:** Molecular Genetics, Technical University Darmstadt, Schnittspahnstr. 10, 64287 Darmstadt, Germany

**Keywords:** direct RNA sequencing, long-read RNA sequencing, nanopore sequencing, transcriptome analysis, African trypanosomes, long noncoding RNAs, bloodstream-stage trypanosomes, insect-stage trypanosomes

## Abstract

Trypanosomatids are single-cell eukaryotic parasites. Unlike higher eukaryotes, they control gene expression post-transcriptionally and not at the level of transcription initiation. This involves all known cellular RNA circuits, from mRNA processing to mRNA decay, to translation, in addition to a large panel of RNA-interacting proteins that modulate mRNA abundance. However, other forms of gene regulation, for example by lncRNAs, cannot be excluded. LncRNAs are poorly studied in trypanosomatids, with only a single lncRNA characterized to date. Furthermore, it is not clear whether the complete inventory of trypanosomatid lncRNAs is known, because of the inherent cDNA-recoding and DNA-amplification limitations of short-read RNA sequencing. Here, we overcome these limitations by using long-read direct RNA sequencing (DRS) on nanopore arrays. We analyze the native RNA pool of the two main lifecycle stages of the African trypanosome *Trypanosoma brucei,* with a special emphasis on the inventory of lncRNAs. We identify 207 previously unknown lncRNAs, 32 of which are stage-specifically expressed. We also present insights into the complexity of the *T. brucei* transcriptome, including alternative transcriptional start and stop sites and potential transcript isoforms, to provide a bias-free understanding of the intricate RNA landscape in *T. brucei*.

## 1. Introduction

Trypanosomatid organisms including the genera *Phytomonas*, *Leishmania,* and *Trypanosoma* are single-cell eukarya that parasitize plants and animals [1], and as such, they are of agricultural, veterinary, and medical importance. Although disease-related objectives have driven most research efforts in the past [2], a significant number of studies have identified a wealth of RNA-centred biochemical phenomena in the different species, including RNA polymerase I-driven transcription of protein-coding genes, *trans*-splicing, RNA editing, ribosomal RNA fragmentation, and mitochondrial tRNA import [3]. This RNA-centricity is a consequence of two specific molecular characteristics of trypanosomatids. First, the organisms exhibit an unusual eukaryotic genome organization. Multiple intronless protein-coding genes are arranged in long head-to-tail tandem arrays, which are transcribed as polycistronic transcripts covering up to 100 coding sequences (CDS). Processing into monocistronic mRNAs occurs via *trans*-splicing of a capped, 39 nt long spliced leader (SL)-RNA to the 5′-end and by 3′-end polyadenylation. Second, transcriptional control of gene expression is, with only a few exceptions, not existent in trypanosomatids. The organisms regulate gene expression post-transcriptionally. This implies that RNA-driven processes, such as splicing, alternative splicing, RNA turnover, and translation, must be able to sense and respond to extrinsic and intrinsic cues to adjust and modulate cellular protein levels. While a comprehensive verification of such a scenario awaits confirmation, the currently accepted view is that changes in the transcriptome are brought about by RNA-binding proteins (RBPs) utilizing a large panel of RNA-interacting proteins [3,4]. RBPs are well-documented components of the sensing and signaling pathways in higher eukaryotes and they execute their function by stabilizing or destabilizing specific mRNAs or other RNA species—reviewed in [5]. Trypanosomatid parasites such as the African trypanosome *T. brucei* rely on a nearly complete inventory of canonical eukaryotic noncoding (nc)RNAs. The panel of small ncRNAs (RNAs < 200 nucleotides (nt) in length) includes the small nuclear (sn)RNA of the *trans*-spliceosome [6], the small nucleolar (sno)RNAs of the ribosomal RNA modification machinery [7,8], as well as transfer (t)RNAs and ribosomal 5S and 5.8S rRNA. Further included are cytosolic Vault (vt)RNA [9], 7SL RNA [10], and telomerase RNA [11] as well as hundreds of guide (g)RNAs, which are a mitochondria-specific class of snRNAs required to edit sequence-deficient mitochondrial transcripts [12,13]. Small interfering (si)RNAs are present in some but not all trypanosomatids and recent experiments identified a new class of anti-sense translational regulators, the so-called TBsRNAs [8]. There is no evidence for canonical microRNAs [3]. Although the cellular roles for most sncRNAs are known, the situation differs drastically for long-noncoding (lnc)RNAs. LncRNAs are defined as noncoding ribonucleic acids with an nt length > 200 nt. They are 5′-capped and 3′-polyadenylated and a growing number have been shown to affect transcription, DNA replication, and repair, as well as cell differentiation—for a review see [14]. About 1500 lncRNAs are annotated in *T. brucei* [15,16]; however, only a single lncRNA (*grumpy*) is functionally characterized. *Grumpy* acts as a precursor for the C/D box-type snoRNA *snoGrumpy*, which is involved in a vital differentiation step in the lifecycle of *T. brucei* [16].

Although the pool of uncharacterized lncRNAs likely contains multiple regulatory lncRNAs, as well as binding targets for regulatory RBPs, it is not clear whether the annotated sequences represent the full complement of lncRNAs in *T. brucei*. This is in part due to the limited number of data but also due to the use of mainly short-read RNA sequencing (RNA-seq). Although RNA-seq has played a dominant role in transcriptome profiling in many organisms, the technique is not without limitations. Firstly, the protocol relies on reverse transcription (RT) to convert RNA into cDNA and on PCR amplification for library construction. Both enzyme treatments have been shown to introduce RT- and amplicon-specific biases [17], especially for sequences with low and high GC contents [18]. Other shortcomings include the misidentification of 3′-ends through internal priming events [19] and ambiguities due to RT-template-switching effects [20,21]. Lastly, the requirement for fragmentation affects preferentially longer transcripts (>100 nt), which makes it difficult to dissect authentic RNA processing sites from artificially introduced cleavage positions, which presents a major obstacle in quantifying and reconstructing the full sequence complexity of the RNA pool [18].

To avoid these drawbacks, here we perform direct RNA sequencing (DRS) of the *T. brucei* transcriptome using the Oxford Nanopore Technologies (ONT) nanopore-based RNA sequencing platform [22]. ONT-DRS is a long-read sequencing technique that is capable of analyzing native RNA strands directly. The technology relies on arrays of membrane-embedded protein pores, through which an electrical current is passed. Aided by a motor protein, individual RNA molecules are fed through the pores and each nucleobase can be identified by the induced shift in ion current [22]. DRS permits the analysis of full-length RNA without the need to amplify and fragment, thereby eliminating all amplification and read-length restrictions. We analyze both major lifecycle stages of the parasite (bloodstream and insect stage) and put special emphasis on cataloging all lncRNAs. We identify 207 previously unknown lncRNAs, 109 of which are expressed in a stage-specific manner. Furthermore, we provide new insights into the complexity of the *T. brucei* transcriptome including poly(A)-tail length variations, alternative splice-acceptor sites (SAS) and polyadenylation sites (PAS), alternative transcriptional start and stop sites, and potential transcript isoforms.

## 2. Materials and Methods

### 2.1. Growth of Trypanosome Cells

*T. brucei brucei* strain Lister 427 [23] was used for all experiments. Bloodstream-stage parasites (MITat serodeme, variant clone MITat1.2) were grown in HMI9 medium [24] supplemented with 10% (*v*/*v*) fetal calf serum (FCS), 0.2 mM 2-mercaptoethanol and 100 U/mL penicillin/streptomycin (Gibco^TM^ Thermo Fisher Scientific, Waltham, MA, USA). Parasites were grown at 37 °C in 95% air and 5% CO_2_ at ≥95% relative humidity. Insect-stage (procyclic) parasites were propagated in SDM-79 medium [25] in the presence of 10% (*v*/*v*) FCS at a temperature of 27 °C. Parasite cell densities were determined by automated cell counting.

### 2.2. Oligodeoxynucleotide Synthesis

Oligodeoxynucleotide primer molecules were synthesized by automated solid phase synthesis using 5′-Dimethoxytrityl-derivatised (**β**-cyanoethyl)-(N,N-diisopropyl)-phosphoramidite monomers. Full-length synthesis products were purified by reverse-phase high-performance liquid chromatography and verified by mass spectrometry. The following DNA sequences were synthesized: 18S_forward: CGGAATGGCACCACAAGAC; 18S_reverse: TGGTAAAGTTCCCCGTGTTGA; β-Tub_forward: TTCCGCACCCTGAAA-CTGA; β-Tub_reverse: TGACGCCGGACACAACAG; nt_4120.1_forward: TGCAAG-TGAAGGCACTGTGT; nt_4120:1_reverse: GCAACAAGCACGGAAGATGG; nt_1759.1_forward: TTGCGTGTGTGTTGGAATGG; nt_1759.1_reverse: ACGACTCAA-AGGAATGCCA; nt_17.3_forward: ACCCGT-AATTTGCGGACTGT; nt_17.3_reverse: TCCCTTCCGTGTATCGTTGT; nt_256.1_forward: GTGTGTATGGGCGTTTCGTG; nt_256.1_reverse: ACATACGCGTAGCCACACAA (18S = 18S ribosomal RNA, β-Tub = β-tubulin, nt = new transcript).

### 2.3. RNA Isolation

Bloodstream-stage trypanosomes were grown to 10^6^ cells/mL and insect-stage parasites to 10^7^ cells/mL. In each case, 5 × 10^8^ cells were pelleted at 3000× *g* and washed in phosphate-buffered saline (PBS: 10 mM Na_2_HPO_4_, 1.8 mM KH_2_PO_4_, pH7.4, 137 mM NaCl, 2.7 mM KCl) supplemented with 20 mM glucose. Cells were shock-frozen in liquid N_2,_ and stored at −20 °C. Total RNA was isolated by guanidinium acid phenol extraction [26]. Polyadenylated RNA (poly(A)-RNA) was isolated from total RNA or crude cell lysates by 2 rounds of affinity chromatography using oligo d(T)_25_-derivatized magnetic beads (NEB). RNA yields were quantified by ultraviolet (UV) spectrophotometry and the integrity of the isolates was electrophoretically analyzed in 2% (*w*/*v*) agarose gels. Poly(A)-enrichment was assessed using qRT-PCR, comparing the ratio of the amount of β-tubulin-specific mRNA over 18S rRNA to that of total RNA using the ΔΔCT method [27].

### 2.4. Synthesis of cDNA and Quantitative Real-Time (qRT)-PCR

Total RNA (200 ng) or poly(A)-enriched RNA (40 ng) was reverse transcribed using 5 U/µL of Superscript IV reverse transcriptase (Thermo Fisher Scientific) in a final volume of 10 µL in the presence of 2.5 µM oligo dT-primer or 1 µM gene-specific reverse (rev) primer. Reaction mixtures were diluted with 10–50 vol. ddH_2_O and 2 µL of diluted cDNA were used in 10 µL qPCR reactions containing the Luna^®^ Universal qPCR Master Mix (NEB) and 0.3 µM of gene-specific forward and reverse primer. PCR reactions were carried out in a StepOnePlus real-time PCR instrument (Applied Biosystems, Waltham, WA, USA) using a modified Fast Run protocol (5 min, 95 °C followed by 30 cycles of 15 s at 95 °C and 30 s at 60 °C). After each run, SYBR green displacement curves were generated (15 s, 95 °C; 1 min, 65 °C followed by a 0.3 °C/15 s temperature rise to 95 °C) with continual fluorescence measurement at 535 nm. PCR products were analyzed by electrophoresis in 10% (*w*/*v*) non-denaturing polyacrylamide gels in 90 mM Tris/B(OH)_3_ pH8.3, 2 mM EDTA buffer (TBE).

### 2.5. Preparation of DRS-Sequencing Libraries and Sequencing

Sequencing libraries were generated from 1 µg poly(A)-enriched RNA following the Oxford Nanopore Technologies (ONT) SQK-RNA002 protocol. Yields were estimated fluorometrically (Qubit dsDNA-assay, Thermo Fisher Scientific) and DRS sequencing was performed on a MinION device in ONT-MinION R9.4 flow cells for 20–24 h. Base calling was carried out after the sequencing runs using Guppy 4.2.2 [28] as part of the ONT-MinKNOW software (v18.03.01). Only reads with a mean *q*-score > 7 were further processed.

### 2.6. Transcript Mapping

A global mapping analysis of the sequenced RNAs was performed using minimap2 (v2.17-r941) [29,30]. For this, sequencing reads were aligned to a composite reference genome consisting of the annotated genome of *T. brucei* TREU927 (v52), the mitochondrial maxicircle genome of *T. brucei* (NCBI M94286.1), and the sequence of the RNA calibration strand (RCS: yeast enolase II, YHR174W) from the Saccharomyces Genome Database (SGD). Only reads from primary alignments were further processed. Please note that strain TREU927 was chosen as a reference genome (see also [31]) because of its higher-quality annotation in comparison to strain TREU427. This includes the no. of unassigned nucleotides, the no. of annotated regions, and especially the annotation of UTRs. While the coding sequences are highly conserved between the two *T. brucei* strains, intergenic, noncoding sequences may be more susceptible to inter-strain variations.

### 2.7. Identification of Full-Length Reads

Full-length reads were identified by taking advantage of the presence of both a 5′-spliced leader (SL)-sequence and a poly(A)-sequence at the 3′-end of every *T. brucei* mRNA. For this, sequencing reads were first mapped to the reference genome. Reads that were uniquely mapped were then appended to the 39 nt SL-sequence, followed by a 2nd mapping step. Reads mapping to at least 10 nt of the SL-sequence were considered 5′-processed. Poly(A)-extensions were analyzed using nanopolish [32,33]. Only reads with both signature motifs were processed further (Appendix A).

### 2.8. Reference-Free Transcript Identification and Identification of lncRNA

Full-length reads from all libraries were combined (N = 835,426) and mapped to the reference genome (TriTryp v52). The alignment file was converted into a bed file using BEDTools v2.26.0 [34,35], and reads on the same strand, overlapping by at least 100 nt, were clustered. Within each cluster, start and end positions were binned over a window of 100 nt. Within each bin, the most frequent positions were considered transcript boundaries. Exons supported by at least 3 reads (N = 10,652 out of a total of 18,887 exons/transcript variants) were considered for the subsequent analysis. lncRNAs were identified from the pool of ≥200 nt transcripts (supported by a minimum of 3 reads) using CPC2 [36] and Lncfinder [37].

### 2.9. Estimation of Transcript Abundance, DGE Analysis, and Statistics

Expression profiles and the analysis of differential gene expression (DGE) were performed using the Bioconductor packages RsubRead [38] and edgeR [39]. *p*-values were adjusted according to Benjamin and Hochberg, and statistical calculations were performed using R (version 4.0.3) and RStudio (version 1.1.453).

## 3. Results and Discussion

### 3.1. RNA Sequencing Libraries—Quality Assessment

We prepared three libraries for each, the bloodstream and insect life-cycle stage of the parasite. Per library, we obtained between 2.8 × 10^5^ and 6.6 × 10^5^ reads (Appendix A), and approximately 90% of all reads were base-called with a median Phred quality (*q*)-score ≥ 7. Both the read quality and the read lengths are comparable for the bloodstream and insect-stage libraries with a median length of 770 ± 57 nt, a median *q*-score of 10.2 ± 0.3, and *p*-values of 0.8 and 0.43. Between 92 and 98% of the reads were mapped to the reference genome of *T. brucei* TREU927 and the mitochondrial (maxicircle) genome of the parasite. The mapping frequencies (Appendix A) are lower for libraries from bloodstream-stage parasites (*p* = 1.5 × 10^−4^), which is due to the high level of variant-surface-glycoprotein (VSG) expression. Three to four percent of all reads map to this transcript. Since the specific VSG sequence is lacking in the annotated TREU927 genome, a higher number of unmapped reads is obtained. Contaminations with ribosomal RNAs are <1.4% and the genome coverage varies between 64% and 73%. A total of 79–84% of the annotated exons (N = 10,728) are detected, with significantly more exons covered in the bloodstream-stage libraries (*p* = 0.0014). As mentioned above, this is due to the presence of reads for VSG and expression-site-associated-gene (ESAG) transcripts. The overall error frequency for all libraries adds up to about 8%. Gene-expression profiles within libraries from either developmental stage correlate with a Spearman rank correlation coefficient (*ρ*) of 0.9. Procyclic and bloodstream-stage libraries correlate with *ρ* = 0.75 (Appendix A).

### 3.2. Identification of Full-Length Transcripts

In contrast to short-read RNA-seq, which requires the assembly of overlapping reads, nanopore-based direct RNA sequencing (DRS) monitors the transcript inventory of cells “directly”. This enables a straightforward mapping of gene boundaries and transcript termini, which are characterized by sharp changes in the genome coverage profiles (Figure 1). As a consequence, DRS allows the identification of novel intergenic and UTR-based transcripts (Figure 1a), it allows the correction of transcript termini, and it enables the detection of transcript isoforms of novel, nonannotated transcripts and pseudogene-derived transcripts (Figure 1b). Intermediates resulting from endonucleolytic cleavage of precursor RNAs such as snoRNA (Figure 1c) or rRNA biogenesis products (Figure 1d) also produce discrete steps in the coverage plots but lack spliced-leader (SL) and poly(A)-tail signatures.

In living cells, full-length transcripts coexist with RNA degradation products, which result from cotranslational mRNA decay, nonsense-mediated mRNA decay, or nucleolytic RNA maturation processes. Since DRS starts from the 3′-end of poly(A)-tailed RNAs, this results in a 3′-end bias. Furthermore, DRS also suffers from a 5′-end ambiguity, because the sequenced RNA strands are released from the nanopore about 10–12 nt before they reach the 5′-terminus [40]. Artificial 5′- and 3′-termini are randomly distributed and generally represented by only single reads. By contrast, true transcript boundaries show higher read counts and are clustered toward the start and end of a transcript (Appendix A).

As stated above, translatable mRNAs and lncRNAs in trypanosomatids are generated by *trans*-splicing and polyadenylation from polycistronic primary transcripts. Thus, mature transcripts are represented as polyadenylated reads with a spliced-leader (SL) sequence at the 5′-end. This unique feature was used to unambiguously identify full-length transcripts. Reads carrying the SL sequence were selected as detailed in Section 2. The error rate of this approach was estimated from the number of putative SL-containing reads for the spike-in control transcript, which is <1%. Due to the 3′-bias, less than 60% of all reads exhibit the 5′-SL-sequence, with significantly fewer 5′-full-length reads (27–37%, *p* = 0.005) in the libraries derived from bloodstream-stage parasites (Appendix A). We obtained a total of 9.7 × 10^5^ reads containing parts of the SL-sequence representing 86,686 putative splice acceptor sites. A total of 80% of the reads that map to annotated exons are localized in the 5′-UTRs, but about 10% map within coding sequences. Examples are shown in Appendix A. They are supported by the proper signature motifs of *trans*-splicing and thus may represent transcript isoforms that encode truncated proteins, novel intragenic lncRNAs, or perhaps reflect misannotations.

Homopolymers in the sequence are characterized as stretches of low variance in the raw-DRS current signal. Homopolymer signals immediately following the signal of the adapter sequence are indicative of a poly(A)-tail, and the extent of this signal can be used to calculate the tail length [33,41]. We used the nanopolish poly(A)-module [32,33] to identify reads containing poly(A)-tails and to access their length. Altogether, 84% of all reads (from all libraries) were identified as polyadenylated (Appendix A). A total of 83% of the reads for the spike-in control and 60% of the reads mapping to the mitochondrial genome were polyadenylated. In addition, the majority of reads that map to rRNA genes were polyadenylated, explaining why some rRNA transcripts have survived the poly(A) enrichment step. However, these poly(A)-tailed rRNAs account for only <0.05% of the cellular rRNA pool and thus likely indicate a low level of background polyadenylation activity of accessible 3′-ends.

The polyadenylated reads could be assigned to 323,153 unique positions in the *T. brucei* genome. About 40% of the polyadenylation sites were assigned to annotated exons, the majority mapping to 3′ UTRs. The median poly(A)-tail length is 96 nt for annotated mRNAs. It is moderately higher for pseudogenic transcripts and lncRNAs (117 nt), but considerably shorter for rRNAs (26 nt) and mitochondrial transcripts (29 nt) (Figure 2a). For the spike-in control, the median tail length matches the reported value of 35 nt [42]. Note that the number of putative polyadenylation or splice-acceptor sites is likely overestimated. The inherent error rate of nanopore sequencing skews the identification of the exact position of polyadenylation sites. Indels immediately upstream of the poly(A)-tails or downstream of putative SAS sites will cause a deviation from the true sites and also lead to a lower read count at the corresponding positions. However, globally the DRS data agree well with the results obtained from short-read RNA-seq (for a comparison, see Appendix A).

### 3.3. Reference-Free Transcript Identification

The selection of full-length reads with both a 5′-SL and a 3′-poly(A)-sequence on the same transcript, enabled us to precisely map all splice variants or transcript isoforms, thereby improving the current transcriptome annotation. For that, we used all full-length reads from all libraries (N = 832,091) and identified 18,406 exons and/or transcript isoforms, many of which are partially overlapping. A total of 30% of the transcripts/isoforms consist of only one read. The maximum number of reads for a single transcript is 6076, which encodes the ribosomal protein L11 (Tb927.9.7620). The transcript length distribution varies from 113 nt to 14,636 nt with a median of 1501 nt. The longest transcript (14,636 nt) covers the complete coding sequence of the trypanosome axoneme component, hydin [43] (Tb927.6.3150). Four other transcripts >10,000 nt contain the complete coding sequences of the paraflagellar rod component 2, the ankyrin-repeat protein, the flagellum attachment zone protein, and the neurobeachin/beige protein (Tb927.6.3670, Tb927.9.15400, Tb927.9.2075, Tb927.10.6150) and comprise 3′- and/or 5′-extensions indicative of incompletely annotated UTRs. All transcripts > 200 nt with more than two reads (N = 10,652) were considered for subsequent analysis. These transcripts/isoforms were assigned to 7332 loci or clusters containing up to 17 different exons or isoforms per transcript. About 50% (N = 5668) of these clusters represent a single exon, while 1664 loci encode 4984 different transcripts/isoforms in total. They either represent splice variants of the same transcription unit or result from alternative or aberrant splicing events, thereby giving rise to split or truncated exons or to di- or poly-cistronic transcripts (see below).

### 3.4. Comparison with Annotated Exons

Of all transcripts with a nucleotide length >200 nt and supported by >2 reads, 818 do not overlap with any annotated exon; 51 overlap by <10% of their nucleotide length. Filtering out exons overlapping with the lncRNAs reported by Guegan et al., 2022 [16] resulted in 355 novel transcripts. However, this approach underestimates the true number, since splice-variants leading to split or truncated mRNAs in addition to pseudogene-derived transcripts are not considered. The annotated *T. brucei* chromosomes contain 9687 cistrons that are expected to be *trans*-spliced and polyadenylated. Of these, 7358 (>10%), 7436 (>5%), or 7496 (>1%) are at least partially covered in our DRS data. Taking into account an uncertainty of ±20 nt in the mapping of transcript boundaries, 2319 observed transcripts match precisely their annotated counterpart. About 10% of the sequenced transcripts represent partial sequences of annotated exons. More than 45% of the annotated exons are completely covered, and include the 5′- and 3′-extensions (median length = 313 nt), which is indicative of incompletely annotated UTRs. Large discrepancies between the annotated and observed length of transcripts can also be attributed to splicing events, which result in truncated RNAs, or to aberrant *trans*-splicing and polyadenylation events that generate long di- or tri-cistronic transcripts. Representative examples of di-cistronic transcripts are shown in Figure 2b and Appendix A. Based on the current annotation, 400 observed exons completely cover two annotated exons, and thus are di-cistronic. Sixteen transcripts are tri-cistronic. All di-and tri-cistronic transcripts carry a 5′-SL-sequence and a 3′-poly(A)-tail with the corresponding genes scattered over all Mbp-size chromosomes of *T. brucei*, with no apparent clustering (Figure 2c). Small ncRNAs were omitted from the analysis, since many of them share a common precursor which per se will cover multiple annotated exons. Transcripts covering >1 exon may be the result of aberrant *trans*-splicing or polyadenylation or simply represent misannotations (for an example see Appendix A). However, in some cases, di-cistronic transcripts account for >20% of all sequencing reads (Figure 2d), which argues against a simple *trans*-splicing or polyadenylation error. In this case, it is more likely that the *trans*-splicing signal is actively suppressed, in order to achieve the desired amount of polycistronic mRNA. Interestingly, we also observed a di-cistronic transcript encoding an LSD1-type zinc finger protein (Tb927.7.4190) and a 3′-UTR-localized lncRNA (Appendix A).

### 3.5. Long Noncoding RNAs

To revise and possibly expand the inventory of lncRNA in *T. brucei* we analyzed all identified full-length transcripts for their coding potential. Two prediction tools that depend on different search criteria were used: CPC2 makes use of the Ficket score, the integrity of the open reading frames, and includes physicochemical properties of sequence-derived peptides [44]. The algorithm is claimed to be species-neutral [36,45]. Lnc_finder from the R package LncFinder [37] relies on intrinsic features of nucleic acid sequences and can be “trained” for any organism of interest. Of 10,652 sequences, 2445 were predicted to be noncoding by CPC2. LncFinder, trained on the TriTrypDBv52 *T. brucei* sequences, identified 1806 putative lncRNAs, and the sequences predicted by both methods (N = 1801) were used for subsequent analysis. Interestingly, both search algorithms also predicted annotated protein-coding sequences as noncoding (18% for CPC2, and 5% for LncFinder) (Appendix A).

#### 3.5.1. Characterisation of Long Noncoding RNAs

The *T. brucei* nuclear genome consists of three chromosome classes: 11 diploid pairs of megabase chromosomes (0.9–5.7 Mbp), intermediate-size chromosomes (300–900 kbp), and minichromosomes (50–100 kbp) [46]. The predicted lncRNA genes cluster at 1382 genomic loci covering all Mbp-size chromosomes (Figure 3a,b and Appendix A). The annotated intermediate-size chromosomes code for an additional four lnc-transcripts. A total of 1139 loci are represented by a single transcript. Of the remaining 243 clusters, 60% contain two and 40% up to seventeen transcripts, which represent splice variants of the same gene and/or overlapping transcripts coded by adjacent loci (Figure 3c,d). The predicted lncRNAs were compared to the annotated *T. brucei* reference genome and the lncRNAs described in [16] (Figure 4a). Sixty-two sequences cover annotated snoRNAs. These transcripts are located at 27 loci, with 12 loci coding for overlapping transcripts (Appendix A). About 45% of the putative lncRNAs overlap with annotated mRNAs and 31 transcripts match the length of the annotated sequences (±20 nt); 75% of these mRNAs code for hypothetical proteins. Seventy-five annotated mRNAs are completely covered by transcripts predicted as non-coding, with 5′ and/or 3′ extensions ranging up to 3100 nt. Out of 93 transcripts overlapping with annotated pseudogenes, 54 sequences are subsequences of pseudogenic transcripts and may represent pseudogene-derived lncRNAs. A total of 17% of the putative lncRNAs (306 sequences) do not overlap with any annotated feature, and thus they formally represent intergenic sequences. However, since the UTR annotation in the reference genome is incomplete, by including the information for transcripts identified from full-length reads, which compensates for incomplete or missing UTRs, 251 transcripts at 452 loci are identified as intergenic. Of the 1491 annotated lncRNAs in the reference genome, 39% are confirmed by our DRS data, and 21% overlap by at least 5% of their nt-length (Figure 4b). About 20% of the annotated lncRNAs are not detected as independent transcripts, but as part of UTRs.

A general comparison of the coding and noncoding transcripts of *T. brucei* for a set of sequence and structure-specific parameters is provided in Appendix A. This includes the nt-length distribution and the nucleotide and dinucleotide content of the RNAs, as well as the thermodynamic stability (ΔG) of the 2D minimum-free-energy (MFE) folds. With a median of 750 nt, noncoding RNAs are shorter than coding RNAs (median = 1600 nt). The molecules have a lower GC content (0.45 versus 0.49) and fold into less-stable 2D structures (ΔG = −0.28 kcal/mol/nt versus −0.32 kcal/mol/nt) (Appendix A).

#### 3.5.2. Novel lncRNAs

The removal from the pool of intergenic noncoding transcripts of all annotated lncRNAs, resulted in 207 novel intergenic loci: 251 transcripts with up to six transcripts per cluster. Examples are shown in Appendix A. A substantial fraction co-localizes with transcription start sites and/or loci coding for variant surface glycoproteins (VSG), expression site-associated genes (ESAG) or genes of the retrotransposon hot spot family (RSH) (Figure 4c–e). The sequences from 41 loci are not unique, with up to four copies, and some have additional counterparts in the genome, which are either not expressed or not detected. These redundant sequences may be arranged as repeats of alternating lncRNA and genes that encode the same protein. Examples are shown in Figure 5. A list of all newly identified lncRNA is provided in Appendix A.

### 3.6. Differential Gene Expression (DGE) between Insect- and Bloodstream-Stage Trypanosomes

Short-read RNA-seq has been widely used to analyze changes in gene expression during the *T. brucei* life cycle [47,48,49]. However, the requirement for reverse transcription and PCR during library preparation has been shown to skew the quantification of transcript levels [50,51]. By contrast, DRS has been shown to accurately enumerate transcript amounts [52,53,54], which tempted us to compare the DRS-derived transcript levels between bloodstream- and insect-stage trypanosomes. Genes showing at least a four-fold change in transcript level with adjusted *p*-values < 0.05 were considered differentially expressed.

Of 11,769 genes, 8355 were included in the analysis based on the criterion of ≥15 reads/gene in all libraries and ≥5 reads in either the procyclic- or bloodstream-stage libraries. A total of 417 genes are up-regulated in bloodstream-stage parasites, compared to procyclic trypanosomes, and 155 genes are down-regulated. As expected, and as shown in Figure 6, the transcripts for stage-specific surface glycoproteins VSG, ISG, and procyclin, as well as expression site-associated genes (ESAG) and procyclin-associated genes (PAG) are over-represented in the panel of differentially expressed genes (*p* = 4.5 × 10^−83^ Fisher’s exact test) and the same holds for members of the retrotransposon-hotspot-protein multigene family (RHS). This is consistent with the observation that many RHS genes or RHS pseudogenes are associated with VSG and ESAG loci in sub-telomeric regions [55,56]. A list of all differentially expressed genes is provided in Appendix A.

Importantly, of the 705 lncRNAs included in the DGE analysis, 109 are differentially expressed (Figure 7a). Only four are up-regulated in procyclic-stage trypanosomes and one of them, the KS17gene_1079a, is flanked by the insect stage-specific procyclin genes Tb927.6.510 and Tb927.6.520. The remaining 105 lncRNAs are up-regulated in bloodstream parasites, and 40% colocalize with VSG, ESAG or RHS genes or with pseudogenes. This suggests a co-regulation of a considerable number of lncRNAs with the expression of stage-specific (surface) proteins. While this might simply reflect a fortuitous co-localization, it is tempting to speculate that the function of one or more of these lncRNAs might be linked to a switch from an inactive to an active VSG-expression site. This represents an attractive working hypothesis awaiting experimental falsification. However, we cannot exclude the fact that the inter-strain differences between *T. brucei* 927 and *T. brucei* 427 might change the picture to some degree. In particular, intergenic, noncoding sequences are likely more susceptible to inter-strain variations. Finally, we experimentally confirmed the differential expression of four of the new lncRNAs (nt_4120.1, nt_256.1, nt_1759.1, nt_17.3) by qRT-PCR, which is shown in Figure 7b. The genomic context of the four lncRNAs, as well as their sequencing coverage profiles for both bloodstream-stage and procyclic-stage trypanosomes, is shown in Appendix A.

### 3.7. The Mitochondrial Transcriptome

The mitochondrial genome in *T. brucei* is organized as a macromolecular assembly of thousands of catenated circular DNA molecules, known as the kinetoplast. Two classes of DNA circles can be distinguished: maxicircles and minicircles. Maxicircles are <25 kbp in size and are present in <50 copies per network. *T. brucei* minicircles have a size of 1 kbp with thousands of copies per kinetoplast. Maxicircles code for key components of the mitochondrial ribosome and the mitochondrial electron-transport and chemiosmosis system; however, some of the protein-coding genes are stored in a cryptic fashion. To generate functional mRNAs, these sequence-deficient transcripts are post-transcriptionally edited, which is detailed in Section 3.8.

Consistent with the observation that RNA polymerase occupies the entire mitochondrial genome [57] we identified sequencing reads covering the entire *T. brucei* maxicircle (Figure 8a). In the coding region, gene boundaries are visible as discrete steps in the coverage profile (Figure 8b). In the variable, i.e., noncoding, maxicircle region, the coverage pattern is rugged, with no defined boundaries (Figure 8c). Transcripts from the noncoding region are only detected for the plus strand. However, based on the data, we cannot decide if the minus strand is not transcribed, or if only transcripts from the plus strand are captured, because of its high A-nt content (Appendix A). Di-cistronic reads are <0.1%, which is in agreement with monocistronic transcription [57]. Steady-state transcript levels, estimated from the sequencing depth (Appendix A), vary over three orders of magnitude, and show the expected stage-specific signatures for edited RNAs (Figure 8b), which are further detailed below.

It should be noted that the absolute values are biased. For example, the 9S and 12S ribosomal RNAs are expected to constitute the most abundant maxicircle transcripts. However, <1% of the reads map to the two rRNAs, possibly because they are 3′-oligouridylated [58] and therefore escape the poly(A)-enrichment step. Similarly, mitochondrial mRNAs have been shown to be 3′-AU tailed [59,60], which, as before, might have caused the depletion of certain transcripts during library preparation. This is likely also reflected in the relative fraction of polyadenylated mitochondrial reads. With a value of 55%, it is considerably lower, compared to the fraction of poly(A)-tailed and *trans*-spliced RNAs (81%). The median length of the poly(A)-tails of all mitochondrial transcripts is about 30 nt, with minor variations between different transcripts. The fraction of polyadenylation and the length of the poly(A)-tails are comparable in both life-cycle stages (Appendix A).

### 3.8. RNA Editing

As outlined above, gene expression of most maxicircle genes relies on a posttranscriptional RNA editing reaction. In the process, sequence-deficient primary transcripts are remodeled into functional mRNAs by inserting and deleting exclusively uridine (U)-nt at specific sites in the pre-mRNAs. For some mRNAs, >50% of the mature sequence are the result of the processing reaction. The sequence information for editing is provided by a trypanosome-specific class of small (40–60 nt long), ncRNAs, termed guide RNAs (gRNAs) [12,13]. In *T. brucei*, gRNAs are mostly minicircle encoded. The molecules hybridize to the primary maxicircle transcripts and control the U-nt insertion/deletion reaction by antiparallel base pairing. At steady-state, >1000 different gRNAs are expressed in the mitochondria of insect-stage African trypanosomes and in the majority of cases, multiple (≤10) gRNAs are needed to fully edit a single pre-mRNA. The reaction proceeds from the 3′-end of the pre-edited mRNA to the 5′-end and as a consequence generates a multitude of partially edited mRNAs, next to fully edited transcripts and never-edited mitochondrial mRNAs.

As shown in Figure 8b, the coverage profiles for never-edited or partially edited mitochondrial RNAs show the described 3′-end bias already observed for nuclear-encoded transcripts (see Cyb mRNA in Figure 8b). However, highly edited transcripts are characterized by an additional distortion towards the 3′-end, which is due to editing. Because of the 3′ to 5′ directionality of the process, only the 5′-regions of partially edited transcripts will align to the genomic sequences, which leads to a gradual drop-off in the DRS profiles towards the 3′-end (see CO3 and A6 transcripts in Figure 8b).

Mapping all reads to the *T. brucei* mitochondrial transcriptome, including all fully edited sequences, was used to monitor the efficiency of the processing reaction (Figure 9). As expected, the data confirm the preferential expression of maxicircle transcripts in insect-stage *T. brucei* and downregulation in bloodstream-stage parasites. The fraction of reads representing fully edited mRNAs ranges from only 0.6% for the CO3 transcript to 25% for the RPS12 mRNA, which suggests that at steady state, fully edited mitochondrial transcripts are turned over rapidly. For genes encoding subunits of respiratory complex I (NADH dehydrogenase,) no completely edited transcripts were detected, in accordance with the findings of Opperdoes and Michels, 2008 [61].

## 4. Conclusions

In this work, we generated a set of genuine RNA sequencing data of the *T. brucei* transcriptome by using direct RNA sequencing on nanopore arrays. This was carried out to circumvent the reverse-transcription and PCR-amplification steps of short-read RNA sequencing, to obtain strand-specific long-read RNA-sequence information. We analyzed the two major stages of the parasite lifecycle using native poly(A)-tailed RNA and generated up to 0.7 × 10^6^ reads per library, with a median read length of 1500 nt and a maximal length > 14,500 nt. A special emphasis was put on the inventory of lncRNAs as potential regulators of gene expression. Of the 1491 lncRNA genes annotated in the reference genome, only 50% were confirmed by DRS. However, we identified 207 previously unknown lncRNAs, 32 of which are stage-specifically expressed with the majority (>96%) up-regulated in bloodstream-stage parasites. Strikingly, 40% of the newly identified lncRNA sequences localize proximal to a bloodstream-stage specific VSG, ESAG, or RHS gene, suggesting a co-regulation scenario perhaps by a similar mechanism. Furthermore, the DRS data provide insights into the complexity of the *T. brucei* transcriptome. This includes 1756 loci, which code for 5203 different transcript isoforms either representing splice variants of the same transcript or alternative or aberrant splicing products resulting in split or truncated exons or di- or poly-cistronic RNAs. In addition, it includes a poly(A)-tail length plasticity between different RNA classes with a median length of 96 nt for mRNAs, 117 nt for lncRNAs, and only 29 nt for mitochondrial mRNAs. Together, the data demonstrate that DRS can provide a more comprehensive and bias-free understanding of the *T. brucei* transcriptome. Provided that the current basecall accuracy and genome-coverage limitations can be overcome [40], it should be possible to gain further insight into the plasticity of RNA processing reactions in *T. brucei*, which should accelerate our understanding of the RNA biology of the parasite.

## Figures and Tables

**Figure 1 genes-14-00610-f001:**
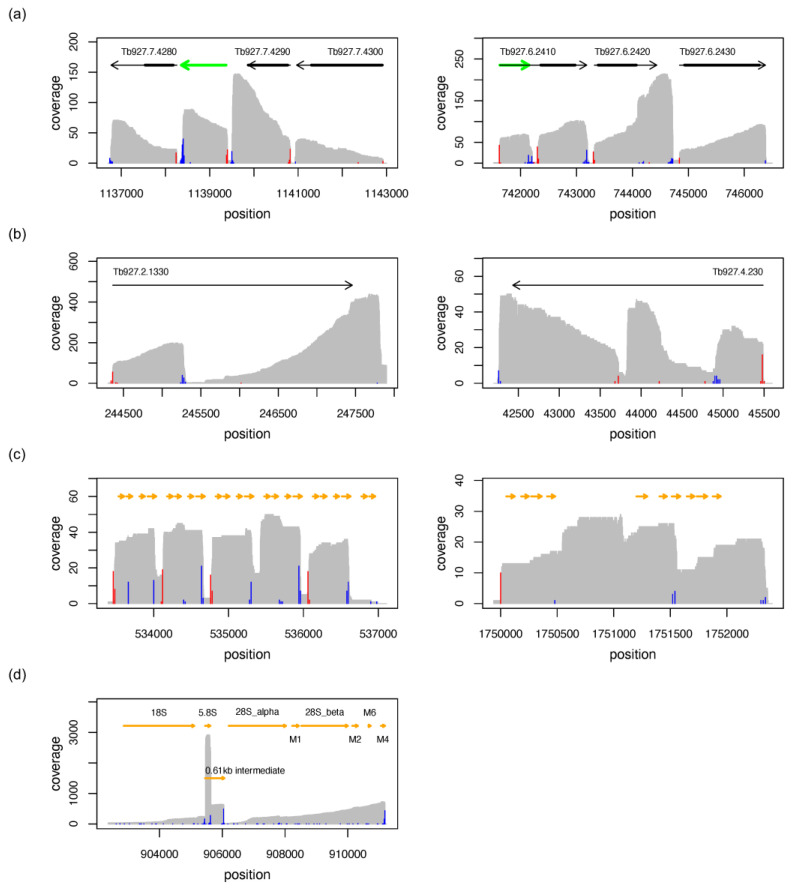
Coverage plots (grey) of representative nanopore-based direct RNA sequencing results. (**a**) Precise mapping of annotated transcript boundaries, correction of annotated UTRs, and detection of novel-intergenic (left) or UTR-based transcripts (right). Grey arrows are annotated genes and thick black lines denote coding sequences. Green arrows are novel transcripts. Splice acceptor sites are in red, and polyadenylation sites are in blue. (**b**) Mapping of pseudogene-derived transcripts. (**c**) Mapping of snoRNA precursors (arrows in yellow). (**d**) Detection of rRNA maturation intermediates (arrows in orange). The data in (**a**,**b**) are derived from bloodstream-stage trypanosomes. The data in (**c**,**d**) are from procyclic-stage parasites. The following genomic regions are presented: (**a**) Tb927_07_v5.1:1136700-1143000 and Tb927_06_v5.1:741520-746500. (**b**) Tb927_02_v5.1:244300-247900 and Tb927_04_v5.1:42200-45600. (**c**) Tb927_08_v5.1:533390-537110 and Tb927_10_v5.1:17499 40-1752400. (**d**) Tb927_03_v5.1:902360-911230.

**Figure 2 genes-14-00610-f002:**
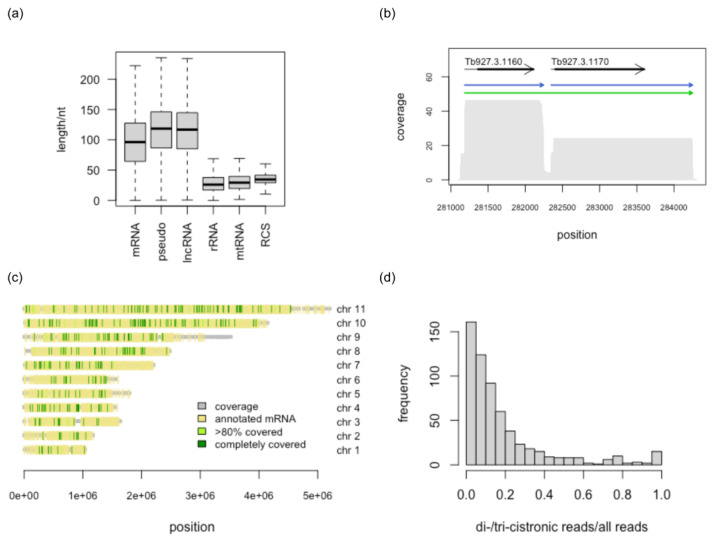
Transcript poly(A)-tail length distribution and characteristics of di- and tri-cistronic transcript reads. (**a**) RNAs are grouped based on the annotated features in TriTrypDB-52_TbruceiTREU927.gff distinguishing messenger (m)RNA, pseudogene-derived transcripts, long noncoding (lnc)RNAs, and ribosomal (r)RNAs, in addition to mitochondrial (mt)RNAs and the RNA-Calibrant-Strand (RCS) used during library preparation. Bold horizontal lines represent the median. (**b**) Full-length read coverage profile of a representative example of a di-cistronic transcript (green arrow) involving the annotated *T. brucei* genes Tb927.3.1160 and Tb927.3.1170. Arrows in blue represent the monocistronic RNAs. Grey arrows are annotated genes and thick black lines specify coding sequences. (**c**) Genomic localization of all di- and tri-cistronic reads on the 11Mbp chromosomes (chr) of *T. brucei*. The color coding is described in the legend. (**d**) Bar graph of the ratio of di- and tri-cistronic transcript reads over all full-length reads that map to annotated exons.

**Figure 3 genes-14-00610-f003:**
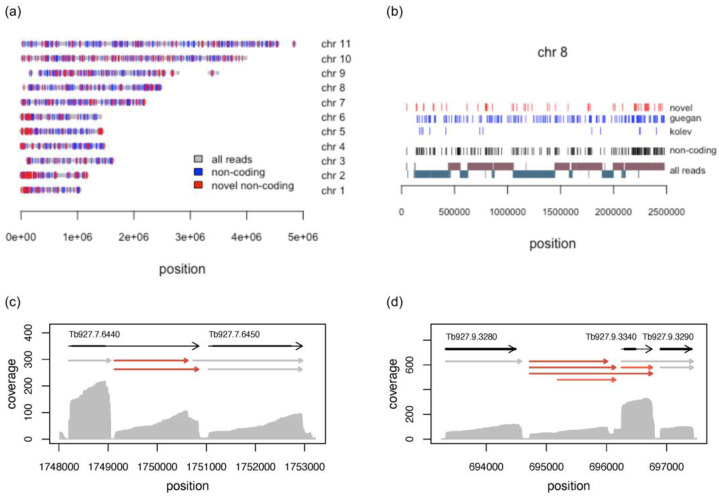
Genomic localization of lncRNA genes. (**a**) Localization of lncRNAs genes on the 11 Mbp chromosomes (chr) of T. brucei. Regions covered by reads, based on all bloodstream- and procyclic-stage libraries are in grey. Known lncRNA genes [15,16] are in blue. Novel lncRNA genes (this study) are in red. (**b**) Detailed view of chromosome 8. The localization of all novel lncRNA genes (red) is separated from the published sequences [15,16] in blue. For comparison, “all reads” are shown in a strand-specific manner. The top line represents the plus strand, and the bottom line the minus strand. (**c**) Clustering of two new lncRNAs (red) as “minor” splice-acceptor or polyadenylation-site variants. (**d**) Complex clustering of novel lncRNAs. Sequencing-coverage plots (grey) from the combined reads of all bloodstream-stage libraries are shown. Grey arrows represent annotated transcripts with coding regions as thick black lines. Grey arrows are predicted coding transcripts and red arrows represent new lncRNAs.

**Figure 4 genes-14-00610-f004:**
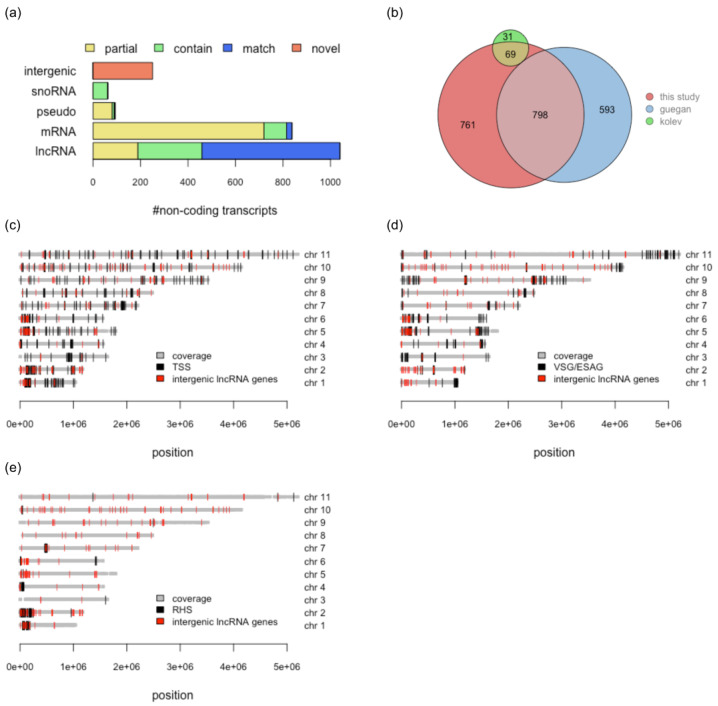
Comparison of noncoding transcripts with annotated and published features. (**a**) Bar graph specifying the number of noncoding transcripts that map to annotated intergenic, snoRNA, pseudogenic (pseudo), mRNA, and lncRNA sequences. The extent of sequence overlap is color-coded. Yellow (partial): >5% overlap. Green (contain): complete coverage including the 5′ and/or 3′ extensions. Blue (match): same length as annotated (±20 nt). Red (novel): novel sequence. Note: the criteria between the different categories are not exclusive. A specific transcript can overlap with more than one annotated feature. Within bars, the numbers are additive. (**b**) Euler plot comparison of the lncRNAs identified in this study (red) with the number of lncRNAs described in [15] (green) and [16] (blue) based on all overlapping sequences (>5%). (**c**) Localization of novel intergenic lncRNA genes (red) on the 11 Mbp chromosomes (chr) of *T. brucei*. Comparison with the positions of variant surface glycoprotein (VSG) genes and expression-site-associated genes (ESAG). (**d**) Comparison with the locations of the retrotransposon-hot-spot gene family (RHS) genes. (**e**) Comparison with transcription start sites (TSS). Bold grey lines: chromosomal regions covered by sequencing reads.

**Figure 5 genes-14-00610-f005:**
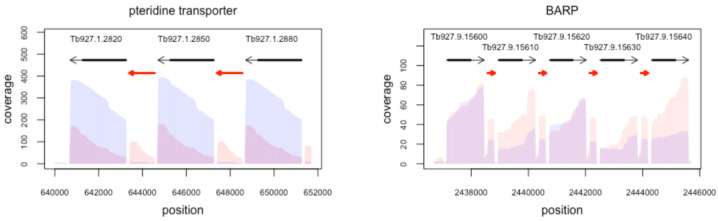
Repetitive lncRNAs. Two examples of genomic regions consisting of repeats of protein-coding genes for identical gene products interrupted by identical lncRNAs. Left: Tandem repeat of pteridine transporter genes on chromosome 1. Right: 3′-domain of the tandem gene array of the bloodstream-alanine-rich protein (BARP) on chromosome 9. Coverage profiles from bloodstream-stage *T. brucei* are in red. Procyclic-stage profiles are in blue. Grey arrows are annotated exons with coding sequences as thick black lines. Red arrows are lncRNAs.

**Figure 6 genes-14-00610-f006:**
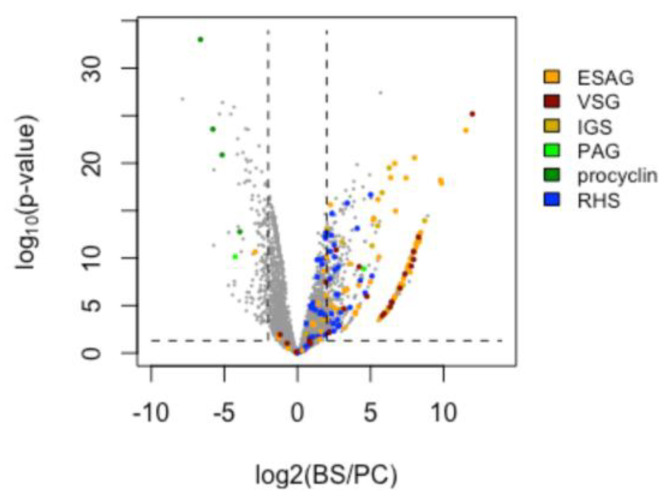
Volcano plot representation of differential gene expression between bloodstream-stage (BS) and procyclic-stage (PC) *T. brucei*. Transcripts for the stage-specific surface proteins procyclin, the variant or invariant surface glycoproteins VSG and ISG, expression site-associated genes (ESAG), procyclin-associated genes (PAG), and retrotransposon hotspot proteins (RHS) are colored as indicated. Dashed vertical lines: fold changes (BS/PC). >4 or <−4. Dashed horizontal line: *p*-value < 0.05.

**Figure 7 genes-14-00610-f007:**
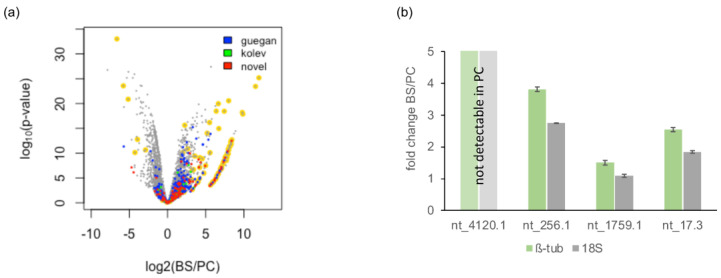
Differential expression of lncRNAs. (**a**) Volcano plot of differential gene expression between bloodstream-stage (BS) and procyclic-stage (PC) *T. brucei*. Novel lncRNA (this study) are in red. LncRNAs described in Guegan et al., 2022, and Kolev et al., 2010 [15,16] are in blue and green. Transcripts for the stage-specific cell-surface proteins procyclin, VSG, ISG and expression site-associated genes (ESAG) are highlighted in yellow. (**b**) Quantitative (q)RT-PCR results of transcript levels of the four novel lncRNAs: nt_4120.1, nt_256.1, nt_1759.1, and nt_17.3. Fold changes (BS/PC) were estimated using the DCT method with either 18S rRNA (18S, grey) or ß-tubulin (ß-tub, green) as controls. Error bars are ±1 SD. For nt_4120.1, no signal was detected in PC trypanosomes.

**Figure 8 genes-14-00610-f008:**
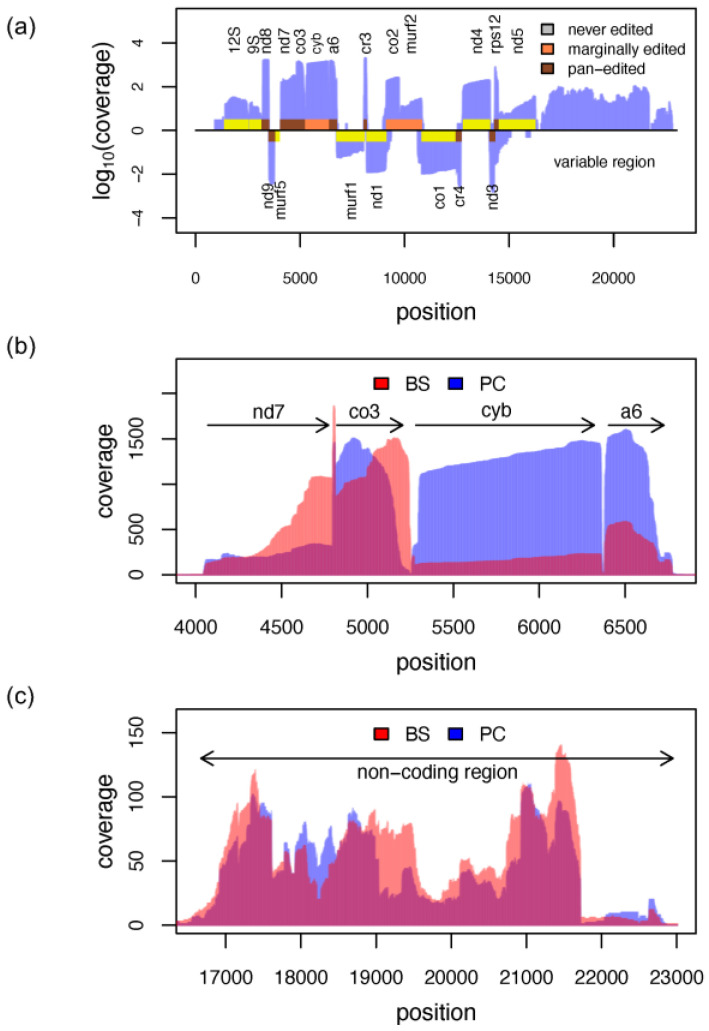
Mitochondrial gene expression. (**a**) Linear map of both strands of the *T. brucei* mitochondrial (maxicircle) genome overlayed with the coverage profile of procyclic-stage trypanosomes (blue). Gene abbreviations are as in [58]. Never-edited genes are in yellow, marginally edited genes in orange, and pan-edited genes in pink. (**b**) Overlay of the coverage profiles from bloodstream-stage (BS, red) and procyclic-stage (PC, blue) trypanosomes in the coding region from ND7 to A6. ND7 = subunit 7 of the NADH dehydrogenase. CO3 = cytochrome oxidase subunit 3. Cyb = cytochrome b. A6 = subunit 6 of the mitochondrial ATPase. (**c**) Overlay of the coverage profiles from bloodstream-stage (BS, red) and procyclic-stage (PC, blue) trypanosomes in the non-coding (variable) region of the *T. brucei* maxicircle genome.

**Figure 9 genes-14-00610-f009:**
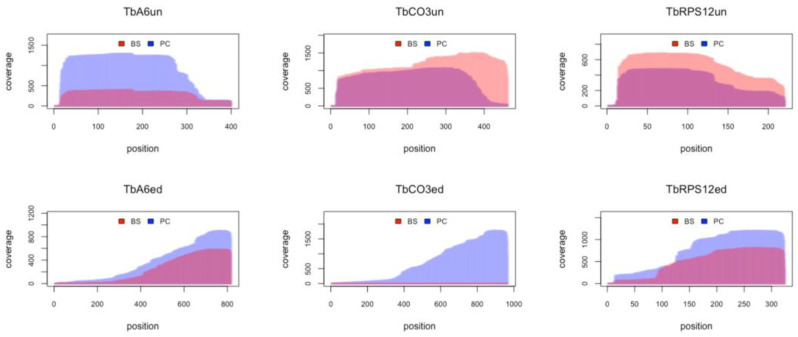
DRS captures the efficiency of mitochondrial RNA editing. Overlay profiles from bloodstream-stage (BS, red) and procyclic-stage (PC, blue) *T. brucei* (Tb) comparing the sequence coverage of unedited (un, top panel) and edited (ed, bottom panel) transcripts of the mitochondrial ATPase subunit 6 (A6), cytochrome oxidase 3 (CO3), and ribosomal protein S12 (RPS12).

## Data Availability

All sequencing data have been deposited in the European Nucleotide Archive (ENA) at EMBL-EBI.

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
