# Peer review of "Nanopore-Based Direct RNA Sequencing of the Trypanosoma brucei Transcriptome Identifies Novel lncRNAs"

_genes, 2023, doi:10.3390/genes14030610_

Round 1

Reviewer 1 Report

The authors use a novel approach to expand the population of putative lncRNAs in T. b. brucei. This analysis is important as T. brucei is largely reliant on post-transcriptional regulation, which is thought to be dependent (at least in part) on RNA binding proteins, and lncRNAs likely have roles to play in this regulation. They identify a panel of novel lncRNAs and present data suggesting that some of these are differentially expressed between the insect and BSF life cycle stages.

The work is largely clearly presented and will provide a useful resource. However, in places the authors should provide additional background information (e.g. on parasite karyotype and the kinetoplast genome) to make the manuscript more accessible to the non-specialist.

My comments are listed by section (in the absence of line numbers) in the order in which the corresponding text occurs in the manuscript.

Abstract

‘with only a single lncRNA characterized to date

Introduction

Para-1. Given the focus on the post-transcriptional regulation of nuclear gene expression, the relevance of editing guide RNAs unclear. These are kinetoplast minicircle-derived and aid in the editing of mitochondrial maxicircle-derived mRNAs. Data on the latter are discussed in the Results section, so should be introduced here and their relevance (or otherwise) to the focus of the paper explained.

Methods

P3: 2.3 RNA isolation. ‘Poly(A)-enrichment was assessed by qRT-PCR’. This typo also occurs elsewhere.

Suggest avoid using the phrase ‘with the help of’ rather say ‘using’ or similar

Results

The work was carried out in the 427-strain. Did you consider mapping against the 427 genome sequences (available at tritrypdb.org), which would likely enable mapping to the expressed VSG? What was the reasoning for not doing this?

While coding sequences are highly conserved between the various T. b. brucei strains, intergenic, non-coding and even lncRNA sequences may be more prone to inter-strain variation. Therefore, mapping to 427 would be expected to enhance the quality of the outputs of this analysis. This is important regarding the accurate identification of putative RNA isoforms derived from alternative processing events, as well as in the accurate mapping of the untranscribed VSG arrays and the sub-telomeric VSG expression sites.

P5: ‘Examples for each case are shown in Figure 1’ – presumably Fig 1a and 1b. The authors should flag in the main text which figure panel illustrates which DRS output.

P7: ‘Also, 83% of the reads mapping to rRNA genes were polyadenylated’ – the authors should comment on this finding, given that in eukaryotes rRNA polyadenylation is thought to be limited and to lead to rRNA degradation (particularly of aberrant transcripts). Is this apparent high level of polyadenylation an artefact of Nanopore sequencing and/or the data processing tool used? This is alluded to later in the paragraph. The authors should clarify this point. They should also emphasise the marked differences in poly-A tail length between the various RNA classes; this is clear in Fig 2 but only touched on in the text.

Fig 2. The final sentence of the legend has broken and is in italics.

P9: 3.5.1 – The authors should briefly outline the chromosomal organisation of T. b. brucei. What does ‘4 annotated small chromosomes of T. brucei’ refer to? The four smallest megabase chromosomes or the parasite’s intermediate size chromosomes (whose number varies between strains and sub-species)?

P10: ‘…the pool of noncoding RNAs is smaller than that of coding RNAs…’

P12: ‘…as well as expression site-associated genes (ESAG, PAG) are over-represented…’ – PAGs are not expressed from expression sites but are associated with procyclin genes. The term 'expression site' exclusively refers to sub-telomeric sites containing ESAGs and a VSG.

P12: ‘40% colocalize with VSG, ESAG or RHS genes or with pseudogenes’ – Presumably lncRNA sequences proximal to VSGs and ESAGs (and RHS genes?) correspond to the untranscribed arrays. Can the authors speculate on the putative function of the lncRNAs in these regions? See also comment above regarding choice of genome for mapping.

P14 (3.7) and P15 (3.8) – As for the parasite karyotype (above), the authors need to briefly outline the important features of the kinetoplast genome (mini and maxicircles), as well as RNA editing and its significance. This should be done in the introduction and relevant details expanded upon in the results section. Can the authors comment on differential expression of maxicircle transcripts, given the differential activity of the parasite’s mitochondrion in BSF and procyclic forms?

P16: ‘Strikingly, nearly every second of the newly identified lncRNA genes’ – Unusual phraseology. I suggest using the actual percentage instead (also on p12, ‘every 2nd lncRNA’). Should ‘genes’ be used in this context, is ‘sequence’ the correct terminology?

Reviewer 2 Report

In this manuscript, Kruse & Göringer use Nanopore technology to survey the repertoire of lncRNAs of bloodstream and procyclic Trypanosoma brucei cells. Although a recent inventory of lncRNAs is available from the work in the Figueiredo's lab, the results presented here will provide the community with a subset of novel, previously unidentified lncRNAs and an in-depth analysis of the catalogue of full-length lncRNAs of trypanosomes.

The manuscript would benefit from the following suggestions, especially those described in point 6 below.

1) In the Introduction, the authors should include RNA polymerase I-driven transcription of protein-coding genes among the RNA-centred biological phenomena described in trypanosomatids.

2) Fig1a, left panel: the gene downstream the novel lncRNA should be represented as well (in the leftmost part of the graph), so the reader can appreciate the 'intergenic' nature of the novel transcript.

3) Polyadenylated rRNA species have been also described in other eukaryotes. Some authors claim that they could represent degradation intermediates. A comparison of rRNA polyadenylation in tryps with that observed in other organisms (length of the poly(A) tail, whether the tails are heteropolymeric, ...) would be illuminating and interesting from an evolutionary perspective.

4) Regarding the 10% of SL-containing reads that map within coding regions: are they supported by 'AG' dinucleotide acceptor sites in the genome?

5) The detection of di- and tri-cistronic mRNAs is intriguing and deserves a bit more detail. I understand that these oligocistrons contain both SL and poly(A) sequences. Is that correct? Could the authors include a table/graphic showing the localization, coverage, ratio of reads of di/tri-cistronic species relative to full-length? And also speculate upon their origin?

6) The authors mention in the main text that "only 599 transcripts (at 452 loci) are identified as intergenic. 153 transcripts in 133 clusters partially overlap with coding sequences of annotated mRNAs. 80 transcripts in 54 clusters map to annotated pseudogenic transcripts, and thus might represent pseudogene-derived lncRNAs. Of the 1491 annotated lncRNAs in the reference genome, 50% are confirmed by our DRS data". These are truly interesting and meaningful results, albeit impractical for the reader unless they are included in Supplementary Table S5. A new column should be added to this Table to indicate if the novel lncRNAs identified in this study i) are intergenic, ii) overlap with coding genes, iii) derive from pseudogenes, iv) are differentially expressed in BF vs PF.
